# The Potential of Strategic Environmental Assessment to Improve Urban Planning in Sierra Leone

**DOI:** 10.3390/ijerph18189454

**Published:** 2021-09-08

**Authors:** Prince T. Mabey, Wei Li, Abu J. Sundufu, Akhtar H. Lashari

**Affiliations:** 1State Key Joint Laboratory of Environment Simulation and Pollution Control, School of Environment, Beijing Normal University, Beijing 100875, China; ptmabey@hotmail.com (P.T.M.); ahussainlashari@gmail.com (A.H.L.); 2Department of Crop Protection, School of Agriculture, Njala University, PMB, Freetown 47235, Sierra Leone; abuj.sundufu@njala.edu.sl

**Keywords:** strategic environmental assessment, urban planning, environmental impacts, sustainable development, West Africa

## Abstract

Strategic Environmental Assessment (SEA) is a proactive and collaborative method for environmental management designed to integrate environmental considerations into decision-making; and it is good for Sierra Leone. To understand whether SEA would be useful in the context of Sierra Leone, the authors interviewed 64 out of 78 experts face to face from March to July 2019. In addition, government policies and regulatory documents on environmental management and sustainable development, published articles served as secondary sources of data. Data were analyzed using descriptive statistics. These Sierra Leonean experts agreed that SEA would be useful for integration and achievement of improved sustainable urban planning strategies. However, the barriers identified to integrating SEA include: not addressing environmental issues during the preparation of policies and programs, insufficient political will, the absence of clear objectives, targets, principles and approaches, overlapping mandates among environmental institutions, and inadequate institutional coordination and non-integrated development framework as barriers to integrating SEA into their work. The study shows that SEA has the potential to have a positive impact on environmental concerns in decision-making, but it would need to be supported by stronger political will, legal frameworks, and improved technical guidance from the policy perspective. Moreover, we propose a conceptual framework for the inclusion of SEA into the urban planning process in Sierra Leone.

## 1. Introduction

In the development planning process, it is increasingly recognized that it is important to identify potential environmental impacts, indicate significant environmental impacts, and try to develop mitigation measures through the generation of alternatives to proposed development. Recognizing the wide range of potential impacts on the environment, a number of planning and assessment procedures have been developed as tools to help achieve more sustainable planning and development. Environmental assessment (EA) is a process for highlighting the possible effects of new development on the environment so that they can be taken into consideration in the decision-making process. In this way, EA is intended to help ensure that development proposals are more sustainable and environmentally sound.

Environmental Impact Assessment (EIA), the first generation of EA, is a systematic process that examines, in advance, the environmental consequences of a proposed development action [1]. Although EIA is now firmly established in the planning process in many countries, some limitations of its application and scope have become evident [2,3]. EIAs are generally applied too late in the decision-making process and often are used to give reason to decisions already taken. Therefore, researchers have realized that there is a need for EA at an earlier point related to policies, plans, and programs (PPPs). Their findings called for the introduction of something other than project level EIA to better address environmental considerations.

Strategic environmental assessment (SEA), the second generation of EA, is interpreted here as the application of environmental assessment at the level of policies, plans and programs. SEA aims to integrate environmental considerations into strategic decision-making [4,5]. It has the potential to make the world a greener and more livable place. However, SEA, as a concept and as a practiced decision-making support tool, is no longer considered a novelty. Its utility in terms of improving the consideration of environmental aspects in policy, plan, and program making and the incorporation of sustainability in the definition of strategies and objectives has been widely developed [6]. Therivel et al. [7] conclude that SEA may be the most direct way of making the sustainability concept operational by providing a comprehensive approach which moves planning processes from their conventional development centered perspective to one that is more environmentally-led. Moreover, SEA has been put forward as an improvement on the existing limited system of EIA. Researchers suggest that SEA can provide a basis for arriving at better-informed decisions at broader strategic levels [8,9]. They claim that SEA can actively integrate environmental concerns into strategic levels of decision making, which ultimately trickles down to the detailed project level.

Sierra Leone has endured several undesired environmental effects as a result of poor environmental planning and lack of adequate environmental considerations in strategic decision-making process. In Sierra Leone, urban planning is a neglected discipline: seldom is it included within the faculty of architecture or engineering and rarely is it an independent discipline. As far as the urban development growth is concerned, Sierra Leone follows the world trend. Half of the population is urban; one third is metropolitan (living in Freetown), and the urban population is expected to double in less than one generation [10]. Over the past decades, the country’s urban expansion has been characterized by leapfrog development, i.e., construction of unbuilt plots not bordering existing development. Nevertheless, with the introduction of the decentralization process which officially started with the Local Government Act of 2004, city/town councils should deal with urban development planning activities in their localities. Regardless of that, an urban planning department up to now does not exist. Planning can significantly influence the livability of cities if its key dimensions (spatial, environmental, socio-cultural and economic) are fully explored. Thus, the scope of planning has entailed shifting focus from conventional planning concerns about land use to concerns about the promotion of sustainable development for cities [11]. The absence of long-term strategic planning coupled with lack of resources (financial and physical) has led to “wild” sprawl and urban growth, because of the ineffective legislations that could prevent urban settlements in conditions that increase the risk of floods, landslides, and other natural disasters especially in the capital Freetown. The capital Freetown is highly urbanized with a significant proportion of its population residing in unregulated/informal and physically-unstable locations. This was evident in the 14 August 2017 mudslide and flooding disaster in the capital that claimed the lives of more than 1000 people.

To date, there are no specific frameworks for taking into account the environmental issues in the preparation of strategic urban development plans. As such, several concurrent problems arise, including environmental degradation, congestion, high population densities, inadequate infrastructure, as well as various social and economic issues. Following specific guidelines, the current law requires an EIA only at project levels. Although there are strong EIA guidelines and regulations, environmental degradation remains a fundamental challenge in developing countries [12], and Sierra Leone is no exception. In addition, EIA has been unable to render these countries with an “environmental sustainability guarantee [13,14]. Therefore, several scholars have acknowledged that SEA can play a role in incorporating environmental factors into decision-making processes for policy plans and programs (PPP), thus contributing to sustainability [15]. Compared to well-developed EIA procedures for specific projects or actions, SEA is an innovation in some developing countries [16]. However, EA of plans and policies as legislation in Sierra Leone does not exist to date.

These existing problems have prompted the assessment of the situation for integrating strategic environmental assessment into the Sierra Leone environmental planning as a method of attaining environmental sustainability in an urban development context.

SEA provides a comprehensive environmental impact assessment of proposed policies, plans, and programs, ensuring that they are discussed adequately at the earliest stage of policy-making [17]. The incorporation of SEA into urban development frameworks as a comprehensive strategy towards sustainable development is essential to mitigate environmental hazards in Sierra Leone. The Sierra Leonean national policy for growth considers that urban planning has to play its role in promoting sustainable development. One approach to this could be the development of a system for integrating SEA into urban planning processes as part of an integrated approach to sustainable development, within the national planning system for Sierra Leone. Despite the mandate of the Local Government Act (2004) that all development and planning processes are passed to the local and municipal governments, environmental management and urban development and planning in Sierra Leone remain extremely concentrated. The institutional framework underlying this centralization is very intricate and obscured, creating a challenging basis for effective planning and management of the environment. The key ministries/agencies and authorities that are directly involved in environmental management and urban planning in Sierra Leone include the Ministry of Lands, Housing and Country Planning (MLHCP), the Sierra Leone Environmental Protection Agency (SLEPA), the Ministry of Works Housing and Infrastructure (MWHI), the Ministry of Local Government and Rural Development (MLGRD), and other stakeholder institutions at the national level. The Law on Decentralization (Local Government Act) in 2004 devolved many urban management functions to the councils, which includes urban planning. However, the councils have not been able to cope with the task, given a lack of urban planning policies and guidelines, a lack of legal and regulatory frameworks compounded by a lack of qualified personnel, as well as lack of resources and working space.

In addition, owing to the lack of clarity in their roles, the management process has been highly fragmented among the various institutions. This complexity makes it very difficult to determine with any certainty the different levels of responsibility, since all the institutions operate at the central government level.

However, being a new phenomenon in the country, the logical first step is to investigate the Sierra Leonean context in order to reveal factors which may promote the potential for SEA, and explore the possible consideration of environmental issues at the strategic level, which has not been looked at before. Then, it is essential to examine and develop its legislative, administrative, and procedural frameworks.

## 2. Materials and Methods

In this study, document analysis and expert interviews were used. With respect to the former, the most relevant documents are internal government papers including relevant policies, guidelines, and legislation, preferably published within the last ten years.

The survey population included the front-line planners and decision-makers who were directly involved in environmental policy making and management, and urban development strategies. In Sierra Leone, at the national level, the main organizations involved in these activities are the MLHCP, the central government body with the responsibility for developing and implementing land use policies, programs, and environmental management; the SLEPA, responsible for the development and implementation of environmental policies, coordination of all environmental management programs and initiation of legislative proposals, standards, and guidelines on the environment in accordance with the 2010 SLEPA Act; and the MLGRD, responsible for providing an effective link between national development priorities and local level development initiatives to bring about more effective service delivery for local people across the country.

In order to identify the survey population and sampling frame, it was necessary to clarify the structure of these institutions, departments, agencies and their responsibilities; and identify those parts most directly involved with environmental assessment and development strategies. This was done by examining the official organization descriptions. After examining the organizational structures and descriptions of the urban and environmental agencies, departments that were directly responsible for urban planning and strategy development or environmental assessment policy and process were identified. The criteria used to identify suitable departments and sections for the survey are listed below:(a).Is the department or section directly responsible for urban development strategies?(b).Is the department or section directly responsible for environmental policy and management?

This survey selected purposive sampling because not every member of staff in each identified agency was suitable for the survey. It was necessary to ensure that selected individuals would be in a position to provide high quality information which this study needed to build interpretive understanding. This study therefore consists of interviewees, which included professional government officials, academic personnel, and environmental experts at national and regional levels that were selected from the identified departments and sections, according to the sampling criteria:(a).Is the staff member directly responsible for urban planning and development strategies?(b).Is the staff member directly responsible for environmental policy formulation and management?

From the sampling criteria stated above, the expert interviews were conducted from March to July 2019. Out of 78 interviewees approached, 64 were willing to be interviewed by face to face, while those unwilling cited their busy schedules as an excuse; the interviewees included professional staff in Ministries, Departments, and Agencies at the planning authorities, which consisted of the government officials, academic personnel, and environmental experts at national and regional levels who are involved in research, and the formulation and implementation of urban development programs (Table 1). These experts are described as senior employees who are responsible for policy formulation at the strategic level in pursuing environmental protection and sustainable development [18,19]. This was done to ascertain the current understanding of environmental impacts and evaluate how SEA could be adopted in the strategic planning process. Each interview began with a non-structured discussion that sought to explore what and how they think about SEA. In order to effectively and comprehensively explore the complex idea of adopting SEA from the selected interviewees and to provide flexibility for contingency questions, the interview moves onto a semi-structured approach, consisting of seven categories of topics and 2–10 questions to be answered using Likert scale, to probe any unclear responses. All experts’ names were kept anonymous in the study and codes were assigned to experts to respect their privacies. The descriptive statistical analyses were used to analyze the data.

## 3. Results

As indicated in Table 1, the majority of the respondents were professional government authorities or public servants (42%), then, environmental experts (23%), academic personnel (19%), and EIA/private consultants (19%) at national and regional levels.

Regarding the advantages of incorporation on SEA into urban planning processes in Sierra Leone, a comparable number of government authorities (29%), professional staff in EIA unit/private consultancy (30%), academic personnel (20%), and environmental experts (21%) strongly agreed that increased public awareness and public participation was one advantage of SEA’s incorporation in the urban planning process in Sierra Leone. Other anticipated advantages, according to government authorities (27%), professional staff in EIA unit/private consultancy (30%), academic personnel (22%), and environmental experts (21%) are attainment of sustainable development; enhanced environmental consideration in decision making (government authorities (28%), professional staff in EIA unit/private consultancy (30%), academic personnel (17%) and environmental experts (25%)), well informed decision—making (government authorities (25%), professional staff in EIA unit/private consultancy (33%), academic personnel (20%) and environmental experts (22%)), and integrated coordination of the planning systems (government authorities (28%), professional staff in EIA unit/private consultancy (31%), academic personnel (23%) and environmental experts (18%) (Figure 1).

However, the government authorities (36%), professional staff in EIA unit/private consultancy (20%), academic personnel (22%) and environmental experts (22%) interviewed, said that, insufficient political will, absence of legal framework (government authorities (30%), professional staff in EIA unit/private consultancy (18%), academic personnel (21%) and environmental experts (31%)), overlapping mandate (government authorities (35%), professional staff in EIA unit/private consultancy (15%), academic personnel (7%) and environmental experts (43%)), lack of SEA capacity (government authorities (30%), professional staff in EIA unit/private consultancy (26%), academic personnel (15%) and environmental experts (29%)) and insufficient methodologies/guidelines (government authorities (39%), professional staff in EIA unit/private consultancy (17%), academic personnel (9%) and environmental experts (35%)) were barriers to integrating SEA into their work (Figure 2). Seventy-five percent (75%) of respondents suggested that the partial integration of SEA in Sierra Leone’s strategic action planning and policy-making will be a more effective and possible approach at present. The partially integrated model is adopted in this study to be the most appropriate model for SEA integration with urban planning process in Sierra Leone (Figure 3; Table 2).

## 4. Discussion

The results of this study found that the advantages of incorporating SEA are increased public awareness and public participation, attainment of sustainable development, enhance environmental consideration in decision making, and advocating well-informed urban development decision-making process, well informed decision-making, and integrated coordination of the planning systems; whilst, the barriers to integrating SEA into urban planning processes in Sierra Leone are insufficient political will, absence of legal framework, overlapping mandate, lack of SEA capacity and insufficient methodologies/guidelines. The partial integration of SEA in Sierra Leone’s strategic action planning and policy-making was considered as the most appropriate model for SEA integration with urban planning process in Sierra Leone.

The above findings correlated with the outcomes mentioned in much environmental research, including integrating environmental concerns more adequately into the process of urban development, improving public participation, and improving accountability and understanding, and achieving sustainability [20,21,22,23].

Environmental issues are considered at project levels, which does not involve the possible evaluation of the long-term environmental outcomes of the places in other development projects. Nevertheless, at the project level, it is mandatory to conduct complete EIA studies under the current legislation, but there is no regulation mandating an Environmental Assessment (EA) for urban plans or programs. A similar scenario in which environmental issues were only considered at the project level was reported in the case of Egypt by Hegazy [24].

According to regulation, the EPASL is eligible for project EIA surveys but has no authority to examine environmental concerns at the strategic levels. Similarly, the Environmental Protection Act 2008 (amended 2010) does not include any phrase indicating the implementation of SEA in the field of urban development. In their article, which compares the SEA-Directive to the SEA-Protocol using the example of spatial planning in Austria, Stoeglehner et al. [25], claimed that environmental goals could be significant in various phases of the planning process: preparation, evaluation, drafting, policy-making, and implementation. However, in Sierra Leone, research on the existing extent of inclusion of environmental concerns into the planning mechanisms and the adoption of urban development programs have shown that environmental concerns are generally limited to only project implementation stage.

Environmental problems in Sierra Leone’s urban development sector pose various challenges in terms of: uncertain procedures, inadequate environmental assessments, low environmental priority, lack of convergence of environmental concerns with development processes, and inadequate coordination between environmental and planning departments. To address these challenges, the introduction of SEA in Sierra Leone is desperately required to guarantee that environmental problems are successfully integrated in the preparation and policy-making phases. The use of proactive environmental evaluation as a way to promote sustainability of planning processes is critical.

### 4.1. Advantages and Constraints of SEA Adoption and Incorporation into the Urban Planning Process

Strategic environmental assessment (SEA) advantages would be feasible if stakeholders and policy-makers become more knowledgeable about environmental problems. Environmental impact evaluation will enable policy-makers to understand the consequences of the actions that are adverse to the environment. Evaluating the potentials of the use of SEA to provide policy-makers with a mechanism to take into consideration the potential environmental consequences of urban development initiatives, is especially relevant. Hence, through SEA, policy-makers were deemed to be consulted at any point of the planning process regarding the costs and benefits of the decisions made. Strengthening public engagement and widening public awareness had the third-highest rate among the respondents. Public involvement would enable the public to learn and become more conscious through an exchange of ideas between citizens and planners. However, some respondents suggested that citizen’s perception might be gathered for discussion purposes, but generally, the public is not involved in the formulation of PPPs for urban planning while some of the respondents recommended more conditional participation depending on the type and consequences of the PPP being measured.

With respect to the principle of SEA to promote sustainable development process, incorporating SEA into Sierra Leone’s planning system would promote sustainability. SEA was widely recognized by the respondents as a valuable component, which provides the justification for more effective policy formulation, planning phase, and program proposals. As a result of PPP, SEA enhances environmental considerations in decision making and advocates for well-informed urban development decision-making process; as adequate incorporation of the planning frameworks would lead to coordination of planning policies among various MDAs.

Adopting SEA as a statutory framework in certain countries is to guarantee that strategic decisions with possibly severe environmental impacts are adequately evaluated, helping to justify it and giving SEA outcomes more significant legal support.

Regarding the key challenges in incorporating SEA into the urban development in Sierra Leone, insufficient political will is the most important constraint to the implementation of SEA; as environmental issues of the nation are portrayed as conflicting with economic development. Therefore, according to them, strong political will with clear environmental initiatives is seen as one of the most significant indicators in the successful implementation of SEA in Sierra Leone.

Interviewees said that the lack of SEA regulations within the state environmental legislative system would prevent the successful implementation of SEA in the planning process in Sierra Leone. Therefore, a legal framework for SEA is important for developing nations like Sierra Leone, because the framework would enhance the feasibility of SEA methods and operations. In addition, the statutory framework should also include specific SEA criteria, guidelines, and obligations.

Moreover, overlapping mandates among environmental institutions are the most essential obstacles which can hinder the successful implementation of SEA, due to the inadequate institutional coordination and non-integrated development framework. For example, the MLHCP, the EPASL, as well as the MLGRD, are currently not able to administer or plan resource usage, in part because of poor coordination, a lack of data and unclear institutional mandates. This was evident in the 14 August 2017 mudslide and flooding disaster in the capital that claimed the lives of more than 1000 people. Thus, the adoption of SEA is necessary so as to ensure that those concerned view themselves as actual players in planning and decision-making systems. Many respondents suggested that the absence of clear objectives, targets, principles, and approaches may be a crucial problem for successful implementation of SEA.

Furthermore, lack of SEA capacity and inadequate technical know-how might hinder the implementation of SEA in the country. In this context, the respondents suggested that capacity building and competent staff are considered to be important components of the successful implementation of SEA.

During the interviews, respondents believed that adequate standards and effective techniques are described as essentially key to SEA implementation. These respondents noted that SEA would be an innovation and therefore, would have procedural and operational challenges in the early stages of its implementation. Hence, to have procedural standards in order to inform stakeholders how SEA is to be conducted is important. In addition, insufficient methodologies/directives would be an obstacle in the implementation of SEA in Sierra Leone. Environmental officials believed that policy-makers would ignore inadequate plans because inaccurate approaches would render SEA ineffective. However, respondents believed that what comprises SEA’s ‘technical requirements and methods’ often depend on the political will, regulatory, and organizational framework [26,27].

### 4.2. The Potential of SEA in the Sierra Leonean Urban Planning Process

This aspect focuses on five fundamental questions as follows;

(a)Does Sierra Leone require a legislative framework for SEA, or it could be adopted voluntarily?(b)If yes, could SEA be incorporated into the current legislation or not?(c)Who should implement SEA?(d)Which approach is most suitable for incorporating SEA into Sierra Leonean urban planning framework?(e)What are the SEA interventions and responsibilities to be introduced in the main decision-making system in urban planning?

With respect to the first question, in the case of Sierra Leone, it will be easier to combine SEA into an acceptable legislative structure for various purposes: legislative criteria for SEA can be conveniently molded to refer to specific fields, as well as the urban planning department; specific legislative structures would be of value to SEA operations in establishing a baseline legislative background or a collection of guidelines, and a statutory foundation could help to establish clear SEA criteria and guidelines that can be more easily applied. The regulatory guidelines should set out the actors and their obligations for the operation of SEA.

Regarding the potential of SEA to be incorporated into current legislation in Sierra Leone, most of the respondents preferred SEA to be incorporated into an established law which might have been motivated by the slow procedural phases of adopting a new regulation in Sierra Leone.

With respect to the question of how SEA might be incorporated into the current legal system and the responsible authority for its adoption, it would be integrated in the current Environment Protection Act 2008 (and its amendments 2010). In addition, SEA could be incorporated into the Terms of Reference (TORs) established by the Ministry of Lands, Housing and Country Planning (MLHCP) as directives for strategic planning. The MLHCP is the legislative entity concerned with the potential design and adoption of urban policies in the nation.

In relation to the body tasked with enforcing SEA, establishing an authority for SEA under the leadership of MLHCP in collaboration with EPA was recommended. This authority will guarantee a degree of accountability in the SEA implementation; and should be accountable for the plan’s evaluation measures, which would approve or oppose both the SEA proposal and the report. The authority should be constituted in collaboration with the EPA to ensure that the early stages of the planning phase take into consideration the SEA principles. This authority shall also be eligible for determining the draft structural proposals on the basis of the results and guidelines of the SEA report. Respondents recommended that the MLHCP initiates and proceed to devise the proposal for urban development. Moreover, the authority proposes the SEA mechanism by examining whether or not the plan requires an SEA. If the revised plan demands SEA task, the authority will select the evaluation team to conduct the SEA, in collaboration with EPA.

With regards to the best approach for integrating SEA into the urban planning system in Sierra Leone, three methods of integration have been identified [28]. The first integration method is used as a basic evaluation technique and totally removed from the planning for strategic intervention. Second, SEA is partly incorporated into the planning of strategic intervention, with minimal chances of exchanging or sharing knowledge. Thirdly, SEA is thoroughly incorporated in the policy planning phase. There was, however, agreement among respondents that a different paradigm will not offer the gains that SEA should be providing. Incorporated planning attracted the strongest endorsement, but the respondents believed it would only be possible if the SEA structure could be developed to enhance institutional collaboration.

Once planning is subject to environmental assessment, a clear and concise connection must be established to ascertain where urban development is possible or not. Hence, strategic planning must be dealt with hierarchically: at state, regional and local levels, from a holistic territorial viewpoint, integrating landscape, cultural, and environmental requirements. In Sierra Leone, environmental assessments should therefore be considered as an integral mechanism for environmental protection and for the inclusion of sustainability requirements into strategic decision-making. The partially integrated model, adopted in this study to be the most appropriate model for SEA integration with urban planning process in Sierra Leone. The approach has the potential to be successful in terms of effective communication, which includes value sharing, institutional coordination, expert integration, and operational integration. The determination of SEA Standards and Indicators with environmental and sustainability concerns will be done considering the following elements of the geodesign framework:Planning Support Systems (PSS) and an iterative process.Information and Communication Technology (ICT) and community participation.Representation Systems, such as GIS and GeoPlanner.

Planning Support Systems (PSS) and an iterative process could be used to analyze spatio-temporal processes to develop approaches and to simulate their impacts on human and environmental domains, Wu [29]. People may learn about the iterative process and develop an ethical and scientific knowledge of sustainability through community engagement using information and communication technology (ICT). Using Representation Systems, such as GIS and GeoPlanner, could aid in the visualization of interdisciplinary scientific information within a geographical environment in order to make complicated science comprehensible to individuals from many professions and to communicate effectively with one another.

All these components and their interconnections ultimately lead to landscape-based sustainability; and will function well if there is excellent coordination within the responsible urban planning institutions including the MLHCP, the EPASL, and the MLGRD. This will ensure sufficient avoidance of specific environmental effects that may emerge, while examining successful remedy or compensation strategies to make urban development consistent with the principles of territorial conservation in Sierra Leone.

Urban development needs a holistic strategy that encompasses all elements of development. Environmental problems must be included in urban planning procedures in order to develop successful sustainable urban policies. Though little attention had been given for evaluating or suggesting changes for environmental performance of urban development, this study looked at the bigger picture to discuss environmental problems on par with economic and social problems and to suggest how environmental problems can be addressed in order to achieve more sustainable patterns of urban development in Sierra Leone.

## 5. Conclusions

Findings from the Sierra Leonean experts suggested that the consideration of environmental concern at the early stages of planning and policy-making, enhancement of public involvement in planning processes, improved coordination among the responsible authorities, and sustainable development are the potential advantages of incorporating SEA into urban planning in Sierra Leone. If Sierra Leone wants to do more SEA, the experts interviewed recommend the following: Improving political support through developing sufficient power to influence different sectors in order to promote SEA implementation.Enhancing environmental policies and objectives in practical terms through sufficient coordination with other planning authorities to inform them with these policies and objectives.Increasing decision makers and planners’ environmental awareness.Developing mandatory provisions related to the adoption of SEA and the enforcement of its results.Improving the negotiation procedures between various authorities involved in SEA.Setting out clear responsibilities of actors involved in SEA process.Developing reliable methodologies and sufficient guidelines.Improving the planners’ knowledge of SEA methods and procedures.

## Figures and Tables

**Figure 1 ijerph-18-09454-f001:**
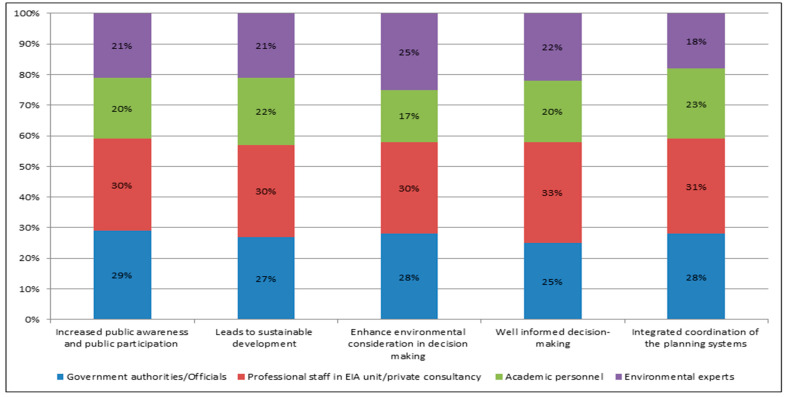
Overall key advantages of SEA’s incorporation in the urban planning process in Sierra Leone.

**Figure 2 ijerph-18-09454-f002:**
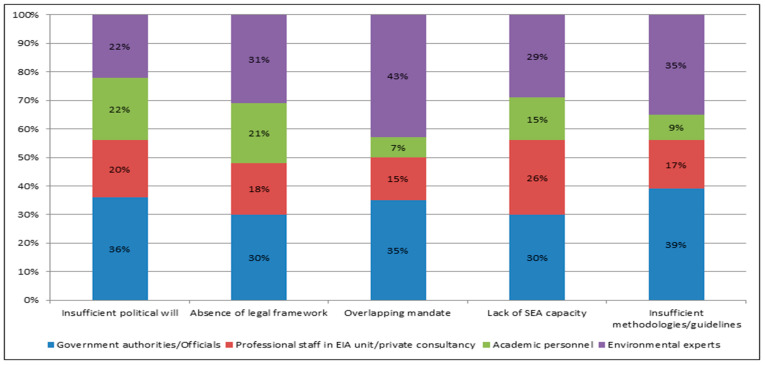
Overall major constraints to SEA SEA’s incorporation in the urban planning process in Sierra Leone.

**Figure 3 ijerph-18-09454-f003:**
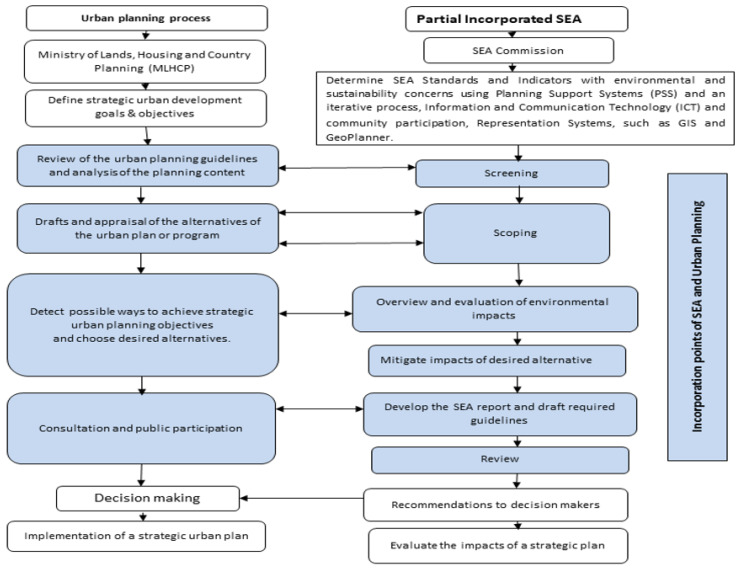
The proposed partial-integration of SEA and urban planning process in Sierra Leone.

**Table 1 ijerph-18-09454-t001:** Demographic profile of respondents.

Institution/Department	Respondent	Number of Respondents	Percent (%)
Ministry of Lands, Housing and Country Planning (MLHCP), Environmental Protection Agency Sierra Leone (EPASL), Ministry of Local Government and Rural Development (MLGRD)	Government authorities	27	42
Environmental research organizations	Environmental experts	15	23
EIA unit/private consultancy	EIA/private consultants	10	16
Colleges and universities	Academic personnel	12	19
Total		64	100

**Table 2 ijerph-18-09454-t002:** The integration between the partial incorporated Strategic Environmental Assessment with urban planning.

Urban Planning Procedure	Urban Planning Task	Incorporated Strategic Environmental Assessment Procedure	Incorporated Strategic Environmental Assessment Task	Planning Question	Sub-Decision Need to Be Made
Review of the urban planning guidelines and analysis of the planning content	Survey and data collection	Screening	Identify the need of Strategic Environmental Assessment	Is Strategic Environmental Assessment necessary for the proposed urban plans, policies, and programs? Are the alternatives for the proposed plans, policies, and programs environmentally sustainable?	Identify the legal requirements of Strategic Environmental Assessment Identify the current environmenal plans, policies and programs.
	Analysis, proposals and evaluation	Scoping	Identify environmental, social, economic and sustainability Issues.	What are the benefits and consequences of action or inaction? What are the possible environmental impacts, of the proposed plans, policies, and programs? Is there any relevant reference?	Identify the assessment requirements Identify the short-term impacts Identify the long-term or cumulated impacts Identify the assessment scale
Drafts and appraisal of the alternatives of the urban plan or program	Identify feasible alternatives		Identify strategic solutions to solve the identified problems.	Are the alternatives for the proposed plans, policies, and programs environmentally sustainable?	Determine the strategic (environmental friendly or sustainable) alternatives such as policy instrument types, transportation modes or sector identification.
Analysis of alternatives	Identify solutions addressing and responding to priority issues	Impact assessment	Overall evaluation	What are the established targets and indicator system for the impacts?	Apply the established assessment approaches.
Formulating the Proposed urban development alternatives	Adjusting selected alternatives	Mitigation measures development	Develop mitigation strategies	What are the drawbacks of the chosen alternatives? Is there any influence on recovery? How might the weaker aspects of the chosen options be mitigated?	Determine the potential mitigation strategies Choosing the most effective mitigation strategies.
Strategic urban development action draft	Reporting	Strategic Environmental Assessment report	Alternatives comparison Synthesizing analysis	What is the effectiveness and implication of each alternative in the priority areas? Which of the proposed measures should be implemented? What are the selected actions’ priorities?	Report the advantages and disadvantages of the alternatives Report the recommendations Report the implementation plan
		Review	Results and recommendations	Is the proposed plans, policies, and programs or other alternatives environmentally feasible or sustainable? What suggestions have been made in response to the study and public consultation?	Determine the findings of the independent and overall evaluations of the proposed plans, policies, and programs, as well as the alternatives. Develop recommendations
Consultation & public participation	Consideration of public participation on the strategic action	Consultation & public participation	Consultation with relevant actors	Which segments of the population will be impacted directly or indirectly? What is the public’s general and specific perception?	Determine the target audience or population. Identify the target audience as well as broader public perception.
Implementation of a strategic urban plan/program.	Check actual performance and adjust measures	Follow-up and Monitoring	Evaluate the impacts of a strategic plan	Who are the responsible stakeholders in the Follow-up team? What are the monitoring and evaluation guidelines and instruments? What are the monitoring indicators? What are the performance and conformance outcomes?	Identify the follow-up team. Identify monitoring and evaluation guidelines and instruments. Evaluate the selected follow-up indicators. Identify uncertainties and unexpected events.

## Data Availability

The data presented in this study are not publicly available but can be provided by the corresponding author in response to a reasonable request.

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
