# Peer review of "The Potential of Strategic Environmental Assessment to Improve Urban Planning in Sierra Leone"

_ijerph, 2021, doi:10.3390/ijerph18189454_

Round 1

Reviewer 1 Report

“Sierra Leone has endured several undesired environmental effects as a result of poor  environmental planning and lack of adequate environmental considerations in strategic 67 decision-making process.” Please provide some example so that the readers that are unfamiliar with the situation in Sierra Leone have a general picture of the situation. The description is too general.

“One approach to this could be the devel-91 opment of a system for integrating SEA into urban planning processes as part of an integrated approach to sustainable development, within the wider national planning system for Sierra Leone. “ Following this statement please provide a short description of the wider national planning system.

The paper addresses to international audience who is not familiar with the urban planning framework and responsibilities in Sierra Leone. Please provide a short introduction on who does what in this field, also in relation to the experts that have been interviewed. Where do they work? What ministries and departments are responsible for certain tasks within the system?

Is the Ministry of Lands, Country Planning and Environment the only authority with responsibilities and tasks in the field?

Figure 2 does not have a good resolution and this makes it difficult to read. Please provide a better image (maybe rewrite the text in the graphic and export it as image).

In the discussion section, the authors state that “overlapping mandates among environmental institutions are the most essential obstacles which can hinder the successful implementation of SEA”, but they do not provide any examples of these mandates. This is related to the previous comment, on providing a summary of the institutions’ mandates and responsibilities

What does MHCP stand for? The manuscript abounds in acronyms, but not all of them are explained (usually at the first occurrence in the text).

Please have the manuscript checked by a native speaker. Some phrases need to be corrected: e.g. What is the plans, policies, and programs contents and influence are? / Is (Are) the proposed plans, policies, and programs or other alternatives environmentally feasible or sustainable?

Author Response

The response file has been uploaded.

Reviewer 2 Report

This is a paper that recommends integrating environmental assessment into the process of making the structure plan of urban land uses at the national / state level instead of by project. This is a really important initiative for environmental protection by early intervention in the development planning. The author has interviewed multiple stakeholders to seek advocacy for the plan, and at the same time identified the possible barriers to the implementations, and thus came out with the suggestion of enforcing partial EA in Sierra Leone. This paper would be a good read for the stakeholders/implementing agencies at the decision-making level, especially if this recommendation of integrating EA is pushed through. Minor comments are in the attached.

Author Response

The response file has been uploaded.

Round 2

Reviewer 1 Report

My suggestions and comments were all satisfactorily addressed. I have no other comments.